# An Improved Skin Lesion Boundary Estimation for Enhanced-Intensity Images Using Hybrid Metaheuristics

**DOI:** 10.3390/diagnostics13071285

**Published:** 2023-03-28

**Authors:** Shairyar Malik, Tallha Akram, Muhammad Awais, Muhammad Attique Khan, Myriam Hadjouni, Hela Elmannai, Areej Alasiry, Mehrez Marzougui, Usman Tariq

**Affiliations:** 1Department of Electrical and Computer Engineering, Wah Campus, COMSATS University Islamabad, Wah Cantt 47040, Pakistan; 2Department of CS, HITEC University, Taxila 47080, Pakistan; 3Department of Computer Sciences, College of Computer and Information Science, Princess Nourah bint Abdulrahman University, P.O. Box 84428, Riyadh 11671, Saudi Arabia; 4Department of Information Technology, College of Computer and Information Science, Princess Nourah bint Abdulrahman University, P.O. Box 84428, Riyadh 11671, Saudi Arabia; 5College of Computer Science, King Khalid University, Abha 61413, Saudi Arabia; 6Management Information System Department, College of Business Administration, Prince Sattam Bin Abdulaziz University, Al-Kharj 16278, Saudi Arabia

**Keywords:** deep learning, machine learning, bat algorithm, artificial bee colony, computer vision, skin lesion segmentation

## Abstract

The demand for the accurate and timely identification of melanoma as a major skin cancer type is increasing daily. Due to the advent of modern tools and computer vision techniques, it has become easier to perform analysis. Skin cancer classification and segmentation techniques require clear lesions segregated from the background for efficient results. Many studies resolve the matter partly. However, there exists plenty of room for new research in this field. Recently, many algorithms have been presented to preprocess skin lesions, aiding the segmentation algorithms to generate efficient outcomes. Nature-inspired algorithms and metaheuristics help to estimate the optimal parameter set in the search space. This research article proposes a hybrid metaheuristic preprocessor, BA-ABC, to improve the quality of images by enhancing their contrast and preserving the brightness. The statistical transformation function, which helps to improve the contrast, is based on a parameter set estimated through the proposed hybrid metaheuristic model for every image in the dataset. For experimentation purposes, we have utilised three publicly available datasets, ISIC-2016, 2017 and 2018. The efficacy of the presented model is validated through some state-of-the-art segmentation algorithms. The visual outcomes of the boundary estimation algorithms and performance matrix validate that the proposed model performs well. The proposed model improves the dice coefficient to 94.6% in the results.

## 1. Background

The uneven development of human skin cells leads to tumours; there are three major classifications of these cancers: squamous cell carcinoma, basal cell carcinoma and melanoma [1]. The stats show that the expected occurrence of skin cancer is 33.33%. The seriousness of this ailment is corroborated by the fact that, worldwide, approximately 20 million cancer cases were reported in 2020, half of which were deadly [2]. A severe type of skin cancer with rapid growth is melanoma, which develops in melanocytes. These cells are highly differentiated, and melanin production is the basic function of melanocytes. With this differentiation operation, the cell’s proliferative potential drops significantly [3].

Classical methods of skin lesion examination comprise naked eye inspection of the lesion’s formation, layout and size measurement. These procedures are less accurate and are time-consuming due to the physical involvement of experts and dermatologists. The timely detection of skin cancer is of the utmost priority, which helps to control the patient death rate. In contrast, artificial intelligent models are progressively resolving these discrepancies [4,5]. New computer-vision-based deep and machine learning methods have been introduced in the past few years. These modern techniques enable practitioners to utilise the aid of a machine to detect and classify these diseases. Therefore, a computer-aided prognosis is essential in achieving enhanced accuracy and effective results [6,7].

Automatic diagnosis is split into three steps: skin lesion preprocessing and boundary estimation, feature extraction/selection and lesion classification based on these features [8,9]. Researchers have used various preprocessing techniques in the past decade to preprocess medical images. These preprocessing methods include artefact removal [7,10,11], colour normalisation [12,13] and contrast stretching [14,15,16]. These preprocessing models are effective but time-consuming and affect the overall algorithm times, including the training and testing durations. As a preprocessing step, the boundary estimation operation is conducted before the classifier to classify the skin lesion images better; this operation segregates the boundaries of skin lesion images. The existence of the segmentation model is essential as it extracts the RoI from nearby background lesions [15,17]. Additionally, this process helps to better recognise skin lesions’ intrinsic clinical features [18]. The exactness of segmentation algorithms is directly related to the effective prognosis of melanoma. Most of the research models generally cornerstone the segmentation methods to enhance the overall precision of the classifier. Researchers have suggested various methods for medical image segmentation models in the literature, such as methods using clustering [19,20,21], deep-learning-based segmentation algorithms [22,23,24], threshold-based methods [16,25] and combined region and statistical methods [10,18]. After extracting the RoI, the portion with the ailment is fed to the classification algorithm for effective outcomes.

Even after the proposal of complex segmentation models, there exists plenty of margin for new research as the process is still an open challenge due to the various databases of skin lesions with added complexity. Extracted features may contain asymmetries such as varying colours, bubbles, hair artefacts, noise and low contrast. These asymmetric features may lead to false classification and affect the accuracy, making the segmentation models more challenging [7,13,14]. At the same time, the segmentation accuracy decreases immensely with changing light, brightness, contrast and levels of distortion in the skin lesion images. This is why a great requirement exists for more efficient methods to cope with these practical issues.

We organise the research article as follows. First, we introduce the topic and its background in Section 1. In Section 2, we mention the intention of the research and our contributions. Then, in the later Section 3, we present related studies, shortcomings and research gaps in the literature. Then, we present the detailed materials and methodology in Section 4. After this, we present our detailed proposed framework in Section 4.2; we divide this section into two parts. The first one contains details about the bat algorithm, and the latter includes the artificial bee colony model. Then, in the later Section 5, we present comparative studies and experimental results. Finally, we conclude the research article in Section 6.

## 2. Problem Statement and Contributions

This study proposes the effects of contrast enhancement on medical image boundary estimation problems. We offer a hybrid metaheuristic method bat algorithm–artificial bee colony, BA-ABC, as a contrast enhancement scheme. In the experimentation process, we utilized three publicly available skin lesion databases: International Skin Imaging Collaboration (ISIC)-2016 [26], 2017 [27] and 2018 [28]. We present some skin lesion samples in Figure 1 from the chosen databases. The major achievements of our research model are (1) the proposal of a hybrid metaheuristic algorithm to estimate image-specific vital parameters for skin lesion contrast stretching, (2) the effective analysis of preprocessing and contrast enhancement on the deep-learning-based boundary estimation models for medical images and (3) the performance evaluation of BA-ABC using comparative studies on the state-of-the-art boundary estimation algorithms.

## 3. Literature Review

The preprocessors are typically useful for attaining improved segmentation results but with compromised processing times. Therefore, preprocessing is considered an essential part of segmentation models. However, no gold-standard model is available for noise removal and contrast stretching in the computer vision and image processing field. Nevertheless, researchers have contributed by proposing many algorithms to resolve this matter. The reduced contrast of database images influences the segmentation results and may reduce the classification accuracy. Researchers presented a model in [16,29] and generated efficient segmentation and classification results to mitigate this issue further. Therefore, they used contrast enhancement as a preprocessing stage to better segregate the RoI from the background for improved segmentation. In addition, they utilised a single RGB image channel to construct a saliency map. The binary image was achieved using the thresholding function. Furthermore, they added a metaheuristic-based particle swarm optimisation (PSO) technique to attain fine-tuned border estimation. Later, they implemented a feature extraction step, followed by another metaheuristic method named genetic algorithm (GA) for feature selection. The benefit of GA is the survival of the fittest chromosome to select features such as outlines, shapes, surfaces and local/global features. Finally, they obtained the classification outcomes by inputting the selected features into a support vector machine (SVM). The variations in image acquisition methods have led to the raised complexity of the segmentation tasks. Due to the absence of a single standard algorithm, researchers have devised various solutions to the stated problem. For example, the authors in [17] introduced a random forest border classifier based on the twelve best segmentation algorithms from the pool of multiple lesion border selection methods. They used a morphological operation for noise reduction, followed by the correction of RoI and background colour features. Seven distinct preprocessing models were performed on input images based on the geodesic active contour (GAC) model. Moreover, the remaining models were based on histogram thresholding algorithms, which utilised entropy and fuzzy logic models.

Researchers on the border detection of skin lesions have offered multifarious models. Additionally, in the last decade, convolutional neural networks (CNN) have attained great popularity and enabled researchers to gain improved segmentation outcomes. However, in practice, downsampled images are fed to these networks due to excessive iterations and parameter calculations; this leads to the loss of features. The authors in article [7] proposed a novel methodology for the efficient segmentation of medical images without a dependency on lesion resolution. The algorithm that they present is a four-step model, which includes hair removal and drawing a bounding box around the lesion; then, they achieve grab-cut segmentation with the Yolov3 deep CNN model and artefact removal with the help of morphological operations as the last step. The increasing number of diagnosed melanoma cases led researchers to propose high-performing segmentation algorithms. However, the existence of artefacts, noise and deviating shapes and arrangements of lesions made this operation more challenging. Therefore, the early diagnosis of this disease has the foremost importance. The researchers in [14] presented a well-performing algorithm for lesion border detection based on the contour model and gradients. Additionally, they used Gaussian distribution patterns to address the varying objects with irregular statistical parameters. The division into various classes is used to consider the probability functionality of mixed intensities; for this, the authors presented methodological modelling for the computation of gradients and attribute extraction. This research uses a publicly available database ($PH^{2}$) for experimentation; Pedro Hispano Hospital Portugal generated this dataset [30]. Automated skin lesion border estimation is essential to initial diagnoses of skin cancer. At the same time, the varying contrast, blurred borders and variously sized lesions make this task more challenging. To cope with this issue, the researchers in [22] present a multi-scale deep learning-based supervised neural network (DSM-Network) model. The research efficiently handled varying-sized lesions using the multi-scaling connectivity block. Additionally, accumulating deep and shallow side output network layers further helped to achieve the task. Finally, they used a preprocessor model named conditional random field (CRF) to purify the contour and enhance the border estimation further. The authors used the publicly available ISBI 2017 [27] and $PH^{2}$ [30] skin cancer databases for extensive experimentation. In the end, they achieved improved results and validated them through performance metrics such as the Jaccard index, dice coefficient, accuracy and sensitivity.

Another article with the same objectives presented fuzzy c-means (FCM) clustering [20]. In addition, the proposed method carried out automated cluster selection using entropy as a histogram attribute. The model catered to the distribution effects of the lesion histogram to segregate lesions from the background. In addition, the Euclidean distance and regional maxima were utilised to achieve the abovementioned task. The normal distribution is detected by first employing the two evident maxima and comparing their distance with some set threshold. To better employ the entropy-based histogram distribution, hue, saturation and value (HSV) colour space conversion is incorporated as an initial step. The authors exploited the value channel for FCM clustering. This model analysed three distinct types of skin lesions: normal, benign and melanoma. This study incorporated a non-dermoscopic image database, University Medical Centre Groningen (UMCG) [31], composed of 170 images with 70 melanoma and 100 benign samples.

A timely skin cancer diagnosis significantly reduces the possible mortality rate. Likewise, effective lesion segmentation leads to improved outcomes of cancer identification. To address this issue while monitoring the posed noise, artefacts and undesired features, the researchers in [21] proposed a method to segment the RoI in medical lesions. The algorithm consists of corrected gamma and keypoint clusters of descriptors. The gamma correction method helped to attain the expected contrast attributes of the input medical image. Furthermore, they extracted the critical attributes from the gamma-adjusted images; clustering them resulted in the efficient segmentation of RoIs. Additionally, they implemented an image smoothing function as a preprocessor to eliminate hair occlusion while blurring the background. While experimenting, the authors utilised publicly available datasets $PH^{2}$ and performed validation on ISIC 2016 to 2019. The authors in [32] used image segmentation using the multi-thresholding technique and enhanced swarm methodology; this incorporates multi-iteration maps and locally estimated escaping operators to extract dermoscopic images’ RoIs. The authors further utilised the two-dimensional entropy-based objective function and incorporated a two-dimensional histogram based on global means to illustrate the sample details. Another study highlighted the impacts of preprocessing on the saliency-based border estimation of medical images [33]. The authors merged colour histogram clustering (CHC) with the Otsu thresholding algorithm to achieve the objective. Additionally, in the preprocessing phase, they contrast-stretched the lesions with contrast-limited adaptive histogram equalisation (CLAHE) and utilised the famous DullRazor for artefact removal. Finally, researchers investigated the impacts of preprocessors on the outcomes of a saliency segmentation strategy based on the association of the CHC model with thresholding based on the Otsu algorithm for medical images. The authors performed experimental analysis on publicly available databases $PH^{2}$ and HAM10000. Another study employed the grasshopper optimisation algorithm (GOA) adn a swarm intelligence (SI) model for lesion segmentation, followed by a feature extraction method named speeded-up robust features (SURF) [34]. The authors classified images from the datasets $PH^{2}$, ISIC-2017 and 2018 into two classes. There is great importance of CNN in medical image-related tasks. At the same time, the poor quality of input images poses a significant challenge in the networks’ training phase, especially in feature extraction parts. Additionally, the background containing noise may become the point of priority for the neural network model, leading to false predicted outcomes. The researchers in [35] proposed a morphological preprocessor established on the geometric shapes of the input images. The study presented a model of geometrical-profile-based h-dome transformation followed by an image histogram correction, able to perform RoI enhancement selectively. As a result, the input to the CNN became a clear RoI with less prominent features in the background area. For experimentation purposes, the authors utilised various datasets, such as X-ray images, breast cancer images, mammography images and skin lesions, to name but a few. The authors in [36] utilised a textural analysis model and grey-channel-based co-occurrence matrices to classify ailments in skin samples. The authors in [37] employed preprocessing techniques such as the inpainting algorithm and top hat filtration. For boundary estimation purposes, they used the grab-cut model. Further, they utilised the inception model for feature extraction, succeeded by a fuzzy classifier. The authors propose a novel preprocessor model in [38] to extract the RoIs of skin datasets. For comparison purposes, they cross-compared the outcomes of SOTA with raw and RoI-extracted samples. The samples with RoIs only resulted in improved responses regarding classification accuracy and execution times.

We have proposed a novel differential evolution–bat algorithm (DE-BA) using hybrid metaheuristic for the preprocessing of skin lesion images [39], and another hybrid technique, differential evolution–artificial bee colony (DE-ABC), is presented in [40]. The proposed algorithms approximate the parameter set using our proposed novel methodologies. This effective preprocessing further helps in achieving effective boundary estimation. Table 1 summarises the preprocessing methods used in some related research.

A thorough review of related studies revealed that medical dataset images might contain hair occlusion, noise and increased similarity between the foreground and background. Additionally, this feature loss leads to over-segmentation and poor classification outcomes. In this way, the role of preprocessing models is necessitated in these conditions. However, plenty of research is proposed in this area; there is still ample space for new work. Primarily, the stated task is performed using a normalisation operation on the last channel of a poor-contrast lesion in the HSI colour space, increasing the RoI, diluting the small amounts of details at the lesion borders and altering the brightness.

As a result, numerous techniques have resulted in enhanced outcomes on medical datasets where these prerequisites were already fulfilled: (1) improved lesions with better segregation from the background, (2) colour normalisation throughout the lesion, (3) noise reduction and artefact removal and contrast enhancement. This led us to propose the preprocessing model.

## 4. Materials and Methodology

### 4.1. Datasets

We have accomplished our goal in this work by utilising three openly available skin lesion databases. Details are as follows.

**ISIC 2016**: This dataset [26] comprises two sets, with 900 samples and 379 samples in the training and testing sets, respectively. This database provides ground truths for all testing and training images. As the present-day classification methods demand various classes, on the contrary, this database only comprises two classes, which fits the requirements of our proposed model.**ISIC 2017**: The set [27] comprises 2600 images, with wo sets, one for training with 2000 lesion samples and the other for testing purposes with 600 image samples, the same as the former dataset. Additionally, the dataset contains separate gold-standard images for both sets. Originally, the challenge for this dataset comprised feature identification for four classes, cancer classification for three classes and tasks such as segmentation.**ISIC 2018**: Researchers produced a database in 2018 [28] comprising two sets with around 1000 testing and 2594 training images in the first two tasks. In the third task, they introduced Ham10,000 with a huge training set of 10,000 with 1512 test images. We have used the train set produced in the first two tasks and divided the set into two parts, the reason being the lack of gold-standard samples for the test set. The division of the database is presented in Table 2.

Several factors impact these skin lesion datasets, including differences in brightness, contrast and shapes; artefacts such as skin types; complicated anatomical structures, and hair occlusions. As a result, epidemiologists may need help in examining noisy images with poor contrast. Advanced segmentation methods, such as artificial intelligence techniques, have been suggested to overcome this challenge.

### 4.2. Proposed Framework

We have shown the model and its connectivity with border estimation algorithms in Figure 2. In the diagram below, we have presented the basic flow of our brightness-preserving contrast-stretching algorithm. The steps show how an input image is transformed to aid the segmentation models. Additionally, we have validated the efficacy of the offered benchmark via the cross-comparison of the border estimation outcomes of original lesions with BA-ABC contrast-corrected samples.

This article proposes a brightness-preserving contrast-stretching technique based on a novel hybrid metaheuristic algorithm named BA-ABC. The colour space of input medical images is RGB; for enhancement purposes, we first converted it to HSI to further extract the intensity channel Υ. As a result, this model yielded a new, enhanced-intensity channel $\Upsilon^{e}$. Figure 3 presents the detailed flow of the contrast-stretching technique.

Our contrast-stretching algorithm keeps the output image dimensions the same as the input image, with *R* number of columns and *P* rows. A generalised formulation for the transformation function of the presented model is detailed below:(1)$$g_{x,y}=T\left\{f_{x,y}\right\},\;\;\;\;\;\forall x\in P,\forall y\in R$$

Optimisation algorithms demand the selection of proper bounds for the parameter set. We have reutilised the fine-tuned bounds for our parameters as assessed in our previous work [40]. The lower bound vector $\psi_{l}$ contains the lower possible values of our parameter set; in contrast, vector $\psi_{u}$ comprises the upper bounds. The metaheuristic algorithms require cross-comparison and choosing the best parameter set greedily. At the same time, the automated functionality of the model requires a well-performing cost function. Practically, there are a number of different functions proposed by researchers such as [41,42]. The cost function that we use comprises three specific implementation measures.
(2)$$C_{F}\left(\Upsilon^{e}\right)=\frac{log\left(log\left(\Upsilon^{s}\right)\right)\times c\_edgels\left(\Upsilon^{e}\right)\times entropy\left(\Upsilon^{e}\right)}{P\times R}$$

The research [43] reveals that an image sample with improved contrast comprises sufficient edgles (neighbouring pixels of the edges), which led to using the sum of the number of edgles and their intensities as a performance measure. Additionally, we used entropy as a third measure. As a result, Equation (Equation 2) reveals the quality of an image based on (a) high entropy of the input image, (b) the raised level of edgels and (c) improved intensity [43]. Therefore, it is a three-step improvement function similar to histogram equalisation, with high entropy values [41]. Our proposed BA-ABC model estimates parameter set values for every image, yielding raised cost values.
(3)$$\Upsilon^{s}=\sum_{x\in P}\sum_{y\in R}\sqrt{{\left(\frac{\partial \Upsilon_{x,y}^{e}}{\partial h}\right)}^{2}+{\left(\frac{\partial \Upsilon_{x,y}^{e}}{\partial v}\right)}^{2}}$$

We have employed the Sobel edge detection filter; we present its horizontal and vertical kernels along with their gradients in Table 3. In addition, the double log in the cost function reduces the impact of edge intensities to overcome the over-contrast stretching issue [39]. Finally, we present the $\Upsilon^{s}$ as a Sobel kernel gradient function in Equation (Equation 3).

Gonzalez introduced statistical methods to enhance the image quality with improved contrast [44]. In 2000, Munteanu [43] improved these methods and offered a single transformation function based on image-specific parameters to achieve the task. The function depended on local area features such as the mean and variance and global attributes such as the number of edge pixels, their intensities and entropy, to name a few. We employed this transformation function in our model and estimated the parameter set using a hybrid of metaheuristic algorithms inspired by natural phenomena. There are two parts in this transformation function; the former includes two parameters $P_{\beta }$ and $P_{\delta }$, and their effects merged with local mean $\Upsilon_{x,y}^{\mu }$ and global mean μ. The advantage of local mean $\Upsilon_{x,y}^{\mu }$ is that it is dependent on the local neighbourhood. The latter part of the transformation function helps to improve the smoothness and brightness preservation based on local mean $\Upsilon_{x,y}^{\mu }$ with parameter $P_{\alpha }$ in the exponent [39].
(4)$$\begin{array}{c} g_{x,y}=\left(P_{\delta }\frac{\mu }{\Upsilon_{x,y}^{\sigma }+P_{\beta }}\right)\times \left[f_{x,y}-P_{\gamma }\times \Upsilon_{x,y}^{\mu }\right]+{\left[\Upsilon_{x,y}^{\mu }\right]}^{P_{\alpha }}\\ \mathrm{Parameter}\;\mathrm{Set}:\to \begin{array}{cccc} P_{\alpha } & P_{\beta } & P_{\gamma } & P_{\delta } \end{array} \end{array}$$

#### 4.2.1. Bat Algorithm

Various parameter sets combine to form a population. Therefore, a randomly initialised population is necessary to start the search operation. Afterwards, the BA starts estimating the optimised value of the parameter set based on the cost function.
(5)$$X_{j}=\psi_{l}+rand\left(0,1\right)\times \left(\psi_{u}-\psi_{l}\right)$$

More than a decade ago, Yang analysed the behaviour of bats and their food-searching techniques and further formulated these search techniques based on the essential bat characteristics, such as velocity, position, pulse emission rate and wavelength [45]. Echolocation is a term used for the discovery and prediction of prey. Averting the determined surroundings using ultrasonic pulses via varying pulses is the main characteristic of bats. By utilising the stated model, frequency is estimated as
(6)$$f_{j}^{\tau }=f_{min}+\phi \left(f_{max}-f_{min}\right)$$

We have used the same parameters as in our study [39]; here, ϕ is an arbitrary number ranging within (0, 1). Additionally, we have formulated the velocity $V_{j}^{\tau }$ and position $X_{j}^{\tau }$ values in the below equations.
(7)$$V_{j}^{\tau }=V_{j}^{\tau -1}+f_{j}\left(X_{j}^{\tau }-X^{*}\right)$$
(8)$$X_{j}^{\tau }=X_{j}^{\tau -1}+V_{j}^{\tau }$$

Updated velocity values are based on the previous velocity, position, present frequency and global best. The present position is then updated too. After these updates, the algorithm selects the best parameter set from the population. Then, it initiates a random walk, similar to bats, to estimate a personal best with the formula
(9)$$X_{new}=X_{old}+\theta L^{\tau },\;\;\;\;\;\theta \in \left[-1,1\right]$$

Bats move randomly in the vicinity of the previously identified optimal location globally. This calculation is based on the global optimum $X^{*}$ and the average loudness $L^{\tau }$, multiplied by an arbitrary factor ϵ. The loudness and pulse rate owned by every bat in the search space is estimated through random numbers [45]. As with the process of temperature cooling in simulated annealing, as the bat approaches its target, its loudness decreases, and its pulse rate $\varphi_{j}^{\tau }$ increases until it reaches $\varphi_{j}^{0}$ [45]. At the time τ+1, the equations for determining the updated pulse rate $\varphi_{j}$ and loudness $L_{i}$ are as given below, where ε and η are constants here.
(10)$$L_{j}^{\tau +1}=\varepsilon L_{j}^{\tau },\;\;\;\;\;\varepsilon \in \left[0,1\right]$$
(11)$$\varphi_{j}^{\tau +1}=\varphi_{j}^{0}\left(1-e^{-\eta \tau }\right),\;\;\;\;\;\eta >0$$

Population fitness is estimated using the cost function, and both values are updated if a better cost is obtained. The best parameter set estimated by the first metaheuristic algorithm is then sent to the contrast modification function defined in Equation (Equation 4), which generates improved contrast of the intensity channel. Intensity channel $\Upsilon^{e}$ with improved contrast is merged with the remaining two modified channels to create the final contrast-enhanced sample. The colour space is transformed back to RGB to obtain a final brightness-preserved contrast-improved lesion. The outcomes showcase the system’s performance through experiments, comparing it with previously successful boundary estimation algorithms.

#### 4.2.2. Artificial Bee Colony

After several iterations, the parameter sets are subjected to the second nature-inspired technique, artificial bee colony (ABC). This approach further optimises the parameters based on distinctive properties. Dervis Karaboga [46] presented ABC in 2005, and in 2007, he published a journal article in the Global Optimisation Journal. Due to its analogy with the social behaviour of animal colonies, it is regarded as a swarm optimisation model. ABC demonstrates better outcomes than other swarm-based strategies, such as fish schools, ant colonies, particle swarms and bird flocks. Work distribution and self-organisation are key features of swarm-based algorithms. The work distribution includes the simultaneous implementation of specific assignments.

Self-association works in light of nearby hunt information rather than the global data, including changes in search reactions, feedback and communications with different worker bees. Simultaneously, worker bee division alludes to the concurrent execution of specific assignments. Unlike BA, in ABC, the objective function cost is not the result achieved through a fitness formula; rather, Equation (Equation 12) is utilised to compute the *fitness_cost*.
(12)$$fitness\_cost=\left\{\begin{array}{cc} 1+|C| & \mathrm{if}\;C<0\\ \frac{1}{1+C} & \mathrm{otherwise} \end{array}\right.$$

The objective function’s cost, *C*, is calculated based on Equation (Equation 2), which remains unchanged from the previous BA algorithm. As ABC is a swarm-intelligent technique, its parameter set comprises food sources. For worker division, all food sources must be used, and new members are randomly assigned using a greedy choice process, which is explained in detail. Food source exploitation with the bees is done in the onlooker, scout and employed bee phases.
(13)$$X_{child}^{u}=X^{u}+\theta \left(X^{u}-X_{part}^{u}\right),\;\;\;\;\;u\in \left[1,D\right]$$

We have utilised Equation (Equation 13) for the mutation procedure when bees are in the onlooker phase. In our scenario, we have a parameter set of size *D*, and θ is a random mutating factor within the range of [−1, 1]. When bees are in the onlooker phase, the mutation is done only on a randomly chosen single partner $X_{part}^{u}$ from the target food source $X^{u}$. The boundaries of the mutated values are adjusted as in the previous model. A strict selection criterion is accommodated to select either parent or child with better fitness_cost using Equation (Equation 14).
(14)$$P_{v,\tau }=\left\{\begin{array}{cc} C_{v,\tau } & \mathrm{if}\;fitness\_cost_{new}>fitness\_cost_{old}\\ P_{v,\tau } & \mathrm{otherwise} \end{array}\right.$$

If a better child is obtained, the trial vector value will be reset to zero. The trial vector tracks the number of failures for each food source, and its value will be increased by one every time a failure occurs. The bees are triggered in the scouting phase when an attempt limit is reached. In the phase with employed bees, all sources have been exploited. The bees in the onlooker phase act similarly to the previous phase, the main distinction being that their likelihood estimate determines the traversal of food sources. This value is calculated using Equation (Equation 15), based on each food source’s fitness and the population’s maximum fitness. Arbitrary choice partners are chosen from the population and food sources to be taken advantage of.
(15)$$Prob_{v,\tau }=0.1+\left(\frac{0.9\times fitness\_cost_{v,\tau }}{max\left(fitness\_cost\right)}\right)$$

The last phase of the ABC algorithm replaces the parameter set that has undergone the highest number of trials with the arbitrary set. The arbitrary set is generated in the same manner as the new population was achieved using Equation (Equation 5), only when the trials exceed the already mentioned threshold. The strict selection is not performed at this stage as the member with trials reaching the threshold is discarded. The newly added member $fitness_{cost}$ is estimated using the same formulae and replaced in the array. At the same time, its trial value is reset to zero. After a set number of iterations or if the termination criterion is met, the global best solution is yielded as the output of our hybrid metaheuristic algorithm. The image-transforming function defined in Equation (Equation 1) comes into play to contrast-stretch the intensity channel. Further steps include fusing the remaining two adjusted channels with the contrast-stretched one to achieve an HSI image. The final step is converting the colour space from HSI to RGB to achieve brightness-preserving, contrast-stretched lesions. After this, the enhanced images can be fed to the boundary estimation algorithm to estimate lesion masks. The following section discusses the outcomes of our algorithm and its validation through the segmentation models based on visual analysis and performance matrix comparison.

## 5. Results

### 5.1. Parameter Setting

We have presented a contrast enhancement technique. The image-transforming function is already discussed in the previous section. The parameter set as the global best is estimated through the hybrid metaheuristic technique to modify the intensity channel contrast. The parameter boundaries are reused as in our previous work [39,40] and are $upper_{bound}=\left[1.6\;0.5\;0.8\;1.5\right],\;lower_{bound}=\left[0\;0\;0\;0.5\right]$. For testing purposes, we have utilised three publicly available skin cancer datasets. The details of the numbers of images selected for training and testing purposes from the chosen datasets are presented in Table 1. We have performed our experimentation on the 1660-GTX GeForce GPU of the NVIDIA company. The presented BA-ABC model was executed in MATLAB 2022b, while the performance of the preprocessor was tested using Python-based boundary estimation algorithms. The first performance matrix used is Jaccard, a similarity index, and the ratio of intersection over union (IoU); it ranges between 0% and 100% and is very sensitive to small changes in the dataset and may behave erroneously when fed with a very small sample set. It is formulated in Equation (Equation 16). The second matrix is the Dice coefficient (F1), the ratio of two times the union over addition. It is formulated in Equation (Equation 17). Here, *FN*, *FP*, *TN* and *TP* stand for false negative, false positive, true negative and true positive, respectively.
(16)$$Jaccard\;\;Index=\frac{TP}{TP+FP+FN}$$
(17)$$Dice\;\;Coefficient=\frac{2TP}{2TP+FP+FN}$$

### 5.2. Boundary Estimation Models

We have used our preprocessed images to validate our proposed model as an input to the boundary estimation algorithms. These boundary estimation algorithms are implemented in Python.

#### 5.2.1. First Boundary Estimation Model—BAT

The proposed preprocessing technique was tested using the boundary-aware transformer (BAT) boundary estimation model [47]. The images were resized to 224 × 224, and a minibatch size of 6 was used. Imagenet was used to pre-train this network’s encoder, and we traversed the database for 300 epochs. After observing 10 consistent epochs, the learning rate was divided by two if an insignificant validation loss arose. In Table 4, we have presented a performance comparison based on the matrix defined above. The comparison reveals that the segmentation model performed well when fed with contrast-enhanced images through our proposed model. At the same time, the size of the database had significant effects on the result of boundary estimation, as better results were achieved with a large database.

The visual comparison of boundary estimation outcomes in Figure 4 reveals our present model BA-ABC to be an efficient technique that improved the outcomes of the segmented mask. The lesion mask generated through enhanced image $\Upsilon^{e}$ closely resembles the gold standard, which is not the case with the original lesion Υ, as presented below.

#### 5.2.2. Second Boundary Estimation Model—CA-Net

We performed a second comparison on another model, the comprehensive attention network (CA-Net) border estimation algorithm, presented in [48]. The results show that the BA-ABC preprocessing technique improved the performance of the CA-Net model, as indicated by the better Jaccard index and Dice coefficient. We used 12 minibatches in our case, with 300 epochs. We utilised adaptive estimation (Adam) as a training algorithm and used 0.0001 as the learning rate value lowered to half after 120 epochs. The parameters remained the same as the original CA-Net paper suggested, and the performance improved slightly in the case of large and diverse datasets, as with the former model (Table 5).

This visual comparison supports the conclusion that BA-ABC enhances the performance of the boundary estimation algorithm, resulting in better-estimated masks than the original images. The enhanced images have better contrast and brightness, leading to a clearer RoI representation and improved segmentation outcomes. The outcomes also reveal that our presented model has the potential to mitigate over-segmentation issues and enhance the outcomes of the segmentation model (Figure 5).

## 6. Conclusions

We have presented BA-ABC as a brightness-preserving local-area contrast-stretching algorithm. In this study, we improved the contrast of skin lesions based on statistical measures such as the local area standard deviation and local mean. The statistical transformation function depended on some vital parameters exclusively estimated through our proposed hybrid metaheuristic algorithm. To validate the efficacy of our presented model, we tested it with some state-of-the-art boundary estimation models. The outcomes revealed that the presented model effectively served as a preprocessor. We utilized publicly available skin lesion datasets ISIC-2016, 2017 and 2018 for experimentation. We plan to extend the range of databases beyond the medical field, such as agriculture, to test our algorithm’s efficacy in the future. We further plan to test the later steps such as feature extraction/selection and then classification using these segmented images. Further, it is planned to explore other metaheuristic methods, such as the Lion Optimisation Algorithm (LOA) presented in [49], or the Red Deer Algorithm (RDA) given in [50].

## Figures and Tables

**Figure 1 diagnostics-13-01285-f001:**
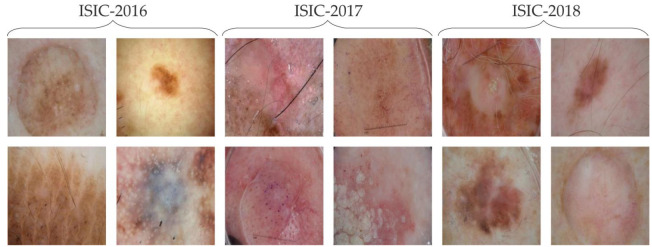
ISIC 2016, 2017 and 2018, four samples each.

**Figure 2 diagnostics-13-01285-f002:**
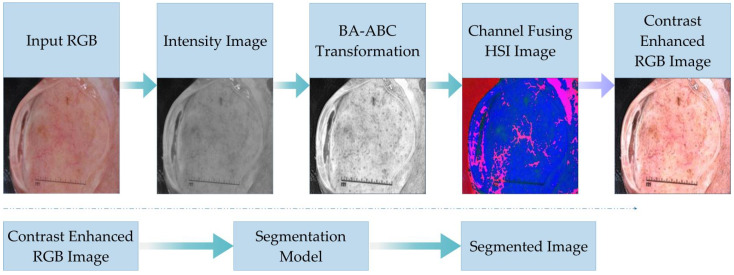
Flow of proposed algorithm.

**Figure 3 diagnostics-13-01285-f003:**
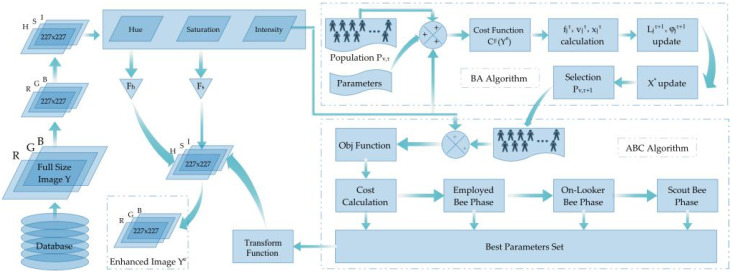
BA-ABC contrast-stretching and transformation flow diagram.

**Figure 4 diagnostics-13-01285-f004:**
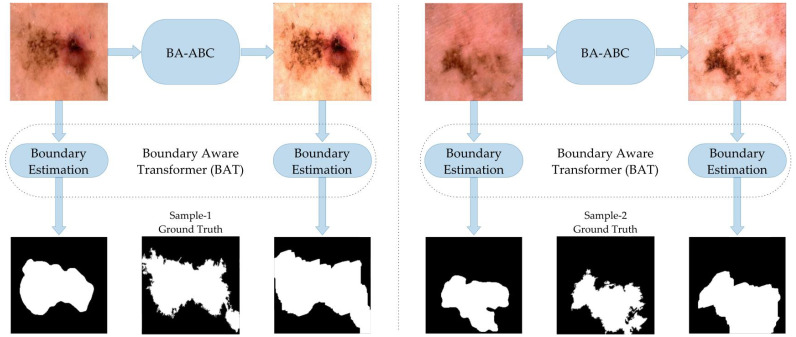
Results with BAT algorithm.

**Figure 5 diagnostics-13-01285-f005:**
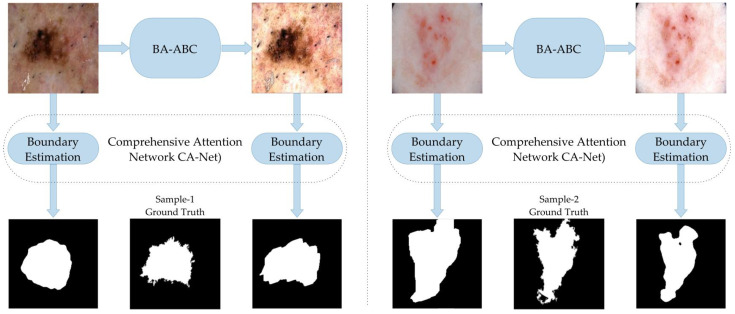
Results with CA-Net model.

**Table 1 diagnostics-13-01285-t001:** Summary of preprocessing used in some boundary estimation procedures.

Paper	Modalities for Preprocessing
[33]	Colour-based histogram adjustment with thresholding based on Otsu algorithm
[34]	Morphological operator-based hair artefact removal and image quality correction using intensity
[35]	Geometrical-profile-based h-dome transformation followed by image histogram correction
[36]	Colour standardisation, normalisation, bottom-hat filtration with discrete Laplacian interpolation
[37]	Top-hat filtration followed by inpainting technique
[38]	Three techniques: Otsu thresholding, K-means clustering and MultiResUNet
[39]	Contrast enhancement with metaheuristic DE-BA
[40]	Contrast enhancement with metaheuristic algorithm DE-ABC

**Table 2 diagnostics-13-01285-t002:** Datasets’ division into test and train sets.

Database	Training Set	Testing Set	Total Images
ISIC-2016	900	379	1279
ISIC-2017	2000	600	2600
ISIC-2018	2076	518	2594

**Table 3 diagnostics-13-01285-t003:** Sobel filter operation.

Sobel Kernel	Gradients
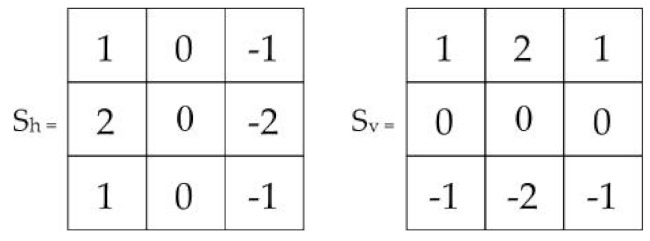	$\frac{\partial \Upsilon_{x,y}^{e}}{\partial h}=S_{h}⊗\Upsilon_{x,y}^{e}$
	$\frac{\partial \Upsilon_{x,y}^{e}}{\partial v}=S_{v}⊗\Upsilon_{x,y}^{e}$
	$\nabla \Upsilon_{x,y}^{e}=mag\left[\frac{\partial \Upsilon_{x,y}^{e}}{\partial h},\frac{\partial \Upsilon_{x,y}^{e}}{\partial v}\right]$

**Table 4 diagnostics-13-01285-t004:** Performance matrix comparison with BAT model.

Algorithm	Dataset		IoU (%)	F1 (%)
	Without	ISIC-2016	85.2	92.0
	Preprocessing	ISIC-2017	86.2	92.6
BAT		ISIC-2018	86.3	92.4
Model	Preprocessed	ISIC-2016	86.9	93.1
	BA-ABC	ISIC-2017	87.0	93.0
		ISIC-2018	87.2	93.2

**Table 5 diagnostics-13-01285-t005:** Performance matrix comparison with CA-Net model.

Algorithm	Dataset		IoU (%)	F1 (%)
	Without	ISIC-2016	85.8	92.4
	Preprocessing	ISIC-2017	89.2	94.3
CA-Net		ISIC-2018	88.3	93.7
Model	Preprocessed	ISIC-2016	87.0	93.0
	BA-ABC	ISIC-2017	88.8	94.1
		ISIC-2018	89.7	94.6

## Data Availability

We have not presented any new dataset from this study. We have utilised some publicly available ISIC datasets. We accessed these datasets via https://challenge.isic-archive.com/data/ (accessed on 27 July 2022).

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
