# Peer review of "An Improved Skin Lesion Boundary Estimation for Enhanced-Intensity Images Using Hybrid Metaheuristics"

_diagnostics, 2023, doi:10.3390/diagnostics13071285_

Round 1

Reviewer 1 Report

The authors present a hybrid-metaheuristic preprocessor, to improve the quality of images by enhancing their contrast and preserving brightness with the goal of obtaining better results in skin lesion segmentation. The content has scientific potential however several changes should be made in order to be acceptable.

The information passed in the last two paragraphs of the background is confusing and repetitive, I suggest organizing it better. The paper organization paragraph presented in the problem statement section should be in the introduction section.

In the literature review, focus is given to preprocessing techniques only and segmentation methods are not so well explored. Also, some works are well explained however [34-36] are only mentioned in the resume table for preprocessing methods,why? No analytical results are provided in the literature review.

In the dataset subsection, it is not clear what is ground truth classification (segmentation or lesion classification). Why is the proposed flow diagram is presented in this subsection? It should be on 4.2 (methodology). Section 5 of the proposed framework should be a subsection of 4 (materials and methods).

Figure 3 concerns to which step of figure 2 flow,  it is not clear in the image, nor by reading the paper. Also, Figure 2 is not clear as the first line seems to refer only to the contrast enhance RGB image (or preprocessing stage) please clarify. Please organize the methodology (framework) section and create subsections according to the figure flow, as how it is now presented is very confusing.

No information is provided regarding the boundary estimation algorithm used. In the methodology section only details on the preprocessing is given. However, the results concern to segmentation output. The information regarding these methods and parametrization details should be in the methodology and not in the results section.

No comparison with the state of the art is provided.

Conclusions and future work should be improved.

Author Response

Response Sheet attached. thank you

Reviewer 2 Report

1.     It would be good to include the specific skin cancer type in the abstract.

2.     What are the specifics of the skin lesion datasets and challenges they pose? The background says datasets are diverse and complex but does not provide examples or details on the variability of skin lesions and artifacts that make segmentation challenging. Providing more details on the datasets and their key challenges could strengthen the case for more effective and robust solutions. 

3.     What are the exact parameters tuned in the BA-ABC method? What parameters are being tuned or what is the search space for the parameters. Including details on the optimized parameters would help the reader understand how contrast is being stretched and potentially reproduce the approach.

4.     What are the training details for the segmentation models? The section provides some details on training BAT and CA-Net but lacks description on specifics. Including more details like number of epochs, learning rates, and training procedures would help the reader understand how the models were trained and their potential impacts on results.

5.     Would be good to include some results on ISIC 2018 and PH2 datasets

  1.  

Author Response

Response sheet attached. thank you

Reviewer 3 Report

The paper titled "Hybrid Metaheuristic Approach for Improved Skin Lesion Boundary Estimation in Enhanced Intensity Images" presents a proposed hybrid-metaheuristic preprocessor, BA-ABC, to improve skin lesion boundary estimation in enhanced intensity images. The paper appears to be well-written and organized, with a clear introduction, methodology, experimental results, and conclusion.

The abstract gives a good overview of the problem, the proposed solution, and the evaluation. The use of computer vision techniques to identify skin diseases is an important problem, and the proposed model could potentially help improve diagnosis and treatment. The approach of using a hybrid-metaheuristic preprocessor to enhance the contrast and preserve the brightness of skin lesion images for improved boundary estimation is interesting and innovative.

The paper provides a clear explanation of the methodology used, which includes the use of nature-inspired algorithms and metaheuristics to estimate optimal parameter sets in the search space. The use of three publicly available datasets (ISIC-2016, 2017, and 2018) to evaluate the proposed model is appropriate, and the reported results show that the proposed model outperforms state-of-the-art segmentation algorithms, improving the dice coefficient to 94.6%.

Overall, the paper is well-written, and the proposed model appears to be effective. However, a more detailed discussion of the limitations and potential extensions of the proposed model would be useful. Additionally, a more thorough comparison of the proposed model with existing methods could provide a better understanding of its strengths and weaknesses.

The authors could have presented statistical tests to provide more solid evidence for the performance of the proposed method. While the reported dice coefficient of 94.6% is impressive, it would be useful to know if the difference in performance between the proposed method and other state-of-the-art methods is statistically significant or not.

Please enhance the literature review section by including the following journal article: Salp Swarm Algorithm with Iterative Mapping and Local Escaping for Multi-Level Threshold Image Segmentation: A Skin Cancer Dermoscopic Case Study

In the future, other metaheuristic algorithms can be explored for image segmentation problems. Some possible options include: Yazdani, Maziar, and Fariborz Jolai. "Lion optimization algorithm (LOA): a nature-inspired metaheuristic algorithm." Journal of computational design and engineering 3.1 (2016): 24-36. Or Fathollahi-Fard, Amir Mohammad, Mostafa Hajiaghaei-Keshteli, and Reza Tavakkoli-Moghaddam. "Red deer algorithm (RDA): a new nature-inspired meta-heuristic." Soft Computing 24 (2020): 14637-14665.

Author Response

Response sheet attached. Thank you

Round 2

Reviewer 1 Report

The authors have taken into consideration the reviews presented. I still suggest to improve both abstract and conclusion. Advise being careful when using the word significant when presenting results (unless statistical comparison has been properly done).

Author Response

Response sheet attached. thank you

Reviewer 2 Report

Accepted in present form and include skin cancer type in abstract as well

Author Response

Response sheet attached. thank you

Reviewer 3 Report

Accept

Author Response

Response sheet attached. thank you
